# Biocytin-Labeling in Whole-Cell Recording: Electrophysiological and Morphological Properties of Pyramidal Neurons in CYLD-Deficient Mice

**DOI:** 10.3390/molecules28104092

**Published:** 2023-05-15

**Authors:** Shuyi Tan, Xiuping Mo, Huihui Qin, Binbin Dong, Jiankui Zhou, Cheng Long, Li Yang

**Affiliations:** 1School of Life Sciences, Guangzhou University, Guangzhou 510006, China; shuyi_tan@m.scnu.edu.cn (S.T.); 2112014047@e.gzhu.edu.cn (X.M.); 2112014048@e.gzhu.edu.cn (H.Q.); zhoujiankui@126.com (J.Z.); 2School of Life Sciences, South China Normal University, Guangzhou 510631, China; 12118014@zju.edu.cn (B.D.); longcheng@m.scnu.edu.cn (C.L.)

**Keywords:** biocytin, whole-cell patch-clamp recording, single-cell labeling, action potential, dendritic spine, CYLD

## Abstract

Biocytin, a chemical compound that is an amide formed from the vitamin biotin and the amino acid L-lysine, has been used as a histological dye to stain nerve cells. Electrophysiological activity and morphology are two key characteristics of neurons, but revealing both the electrophysiological and morphological properties of the same neuron is challenging. This article introduces a detailed and easy-to-operate procedure for single-cell labeling in combination with whole-cell patch-clamp recording. Using a recording electrode filled with a biocytin-containing internal solution, we demonstrate the electrophysiological and morphological characteristics of pyramidal (PNs), medial spiny (MSNs) and parvalbumin neurons (PVs) in brain slices, where the electrophysiological and morphological properties of the same individual cell are elucidated. We first introduce a protocol for whole-cell patch-clamp recording in various neurons, coupled with the intracellular diffusion of biocytin delivered by the glass capillary of the recording electrode, followed by a post hoc procedure to reveal the architecture and morphology of biocytin-labeled neurons. An analysis of action potentials (APs) and neuronal morphology, including the dendritic length, number of intersections, and spine density of biocytin-labeled neurons, were performed using ClampFit and Fiji Image (ImageJ), respectively. Next, to take advantage of the techniques introduced above, we uncovered defects in the APs and the dendritic spines of PNs in the primary motor cortex (M1) of deubiquitinase cylindromatosis (CYLD) knock-out (*Cyld*^−/−^) mice. In summary, this article provides a detailed methodology for revealing the morphology as well as the electrophysiological activity of a single neuron that will have many applications in neurobiology.

## 1. Introduction

Patch-clamp recording is a powerful tool for the study of excitable cells, particularly neurons, allowing for electrical measurements of currents resulting from the activation of ion channels located on neuronal membrane surfaces. An action potential (AP) occurs when the cell membrane potential rapidly rises and falls; APs play a central role in neuron-to-neuron communication. An AP assists the propagation of signals along the neuron’s axon towards the synaptic boutons, and the resulting neurotransmitter signals released from the synapses are delivered to other neurons [1]. In addition, the dendritic spine of a synapse is an important factor in determining neuronal electrophysiological properties [2,3,4]. Simultaneous observation of both the electrical activity of an individual neuron and its morphology, which would allow for the correlation of morphology and function in a single neuron and thus the elucidation of neuronal phenotype, represents a major technical challenge in the field of neuronal electrophysiology. The introduction of dyes into cells via iontophoresis, pressure injection, and diffusion from micropipettes is a powerful technique in neuroscience with many applications; it is particularly useful for cell identification following electrophysiological recordings, allowing for the delineation of the cellular architecture corresponding to the electrical activities of the recorded neuron [5]. After breaking into a giga-ohm seal with ramping suction via a glass capillary recording electrode, the so-called whole-cell mode is achieved. In principle, after establishing whole-cell access in this way, a dye loaded into the electrode solution should diffuse freely into the cell, allowing for subsequent anatomical reconstructions and the morphological analysis of the recorded cell [6]. Biocytin is a conjugate of D-biotin and L-lysine that results from the formal condensation of the carboxylic acid group of biotin with the N(6)-amino group of L-lysine [7]. Due to its high affinity for binding to avidin, biocytin can be visualized by using avidin-conjugated markers [8]. Thus, biocytin has been used for intracellular filling in living neurons to visualize the morphology of dendritic arborization [9]. In practice, however, simultaneous single-cell labeling and whole-cell patch clamp recording is challenging due to the complexity of the procedures involved. Failures either in achieving a tight seal of the electrode during whole-cell mode, in dye loading, or in post hoc immunocytochemistry can result in the unsuccessful pairing of electrophysiological activity with morphology in the same neuron.

Cylindromatosis (CYLD) is critically involved in regulating synaptic function and neuronal morphology via its deubiquitinating function, which specifically cleaves lysine 63- and methionine 1-linked polyubiquitin chains in a wide variety of biological processes [10,11,12,13]. CYLD modulates hippocampal plasticity, and a deficiency in CYLD results in a decrease in dendrite length associated with spine loss in hippocampal pyramidal neurons; a lack of CYLD also reduces the neurite outgrowth of auditory nerve cells [14,15]. The striatum receives projections from the pyramidal neurons (PNs) of the primary motor cortex (M1), and this corticostriatal pathway plays an important role in motor learning and action planning [16,17]. However, whether CYLD deficiency affects the electrophysiological activity and morphology of PNs in the M1 region has never been reported. In the present study, we uncovered changes in whole-cell APs and dendritic spines in PNs of the M1 region of *Cyld*^−/−^ mice by introducing an easy-to-follow methodology involving single-cell labeling by biocytin of whole-cell recorded neurons. This study indicates a critical role for CYLD in PN neuronal activity and morphology.

## 2. Results

### 2.1. Whole-Cell Recording and Biocytin Labeling Procedure

A flowchart of the whole-cell recording/biocytin-labeling technique is shown in Figure 1A. Using the PV-Cre mouse line and the recombinant viral construct AAV-hSyn-DIO-EGFP [18], a mouse model with GFP expression in its striatal PV interneurons was generated (Figure 1B). PV interneurons and MSNs of the striatum and PNs of the M1 region of the cortex were recorded in this study (Figure 1C). The PV interneurons were recognized by their expression of GFP, while the PNs and MSNs were initially identified by their morphology using IR-DIC microscopy (Figure 1D), which was then confirmed later by a morphological analysis of the biocytin signals. The whole-cell recording was conducted, and the APs of each recorded neuron were captured in current-clamp mode using an internal solution containing biocytin (Figure 1D). Having captured the APs of the neurons loaded with biocytin, the brain slice was fixed and then incubated with Streptavidin Alexa 594 (red) or Streptavidin Alexa 488 (green) for 24 h, after which images were obtained by confocal microscopy (Figure 1E).

### 2.2. Whole-Cell Recording and Biocytin Labeling of PV Interneurons

GABAergic PV interneurons constitute ~1% of the striatal neuronal population [19], receiving excitatory input from the cortex and exerting a potent feedforward inhibition onto nearby MSNs. Though few in number, PV interneurons are thought to control the integration of cortical information by regulating MSNs, which in turn regulate the activity of striatal microcircuits [20]. Acute brain slices were used to record the APs of the GFP-expressing PV interneurons. In line with a previous report [20], we observed that striatal PV interneurons are fast-spiking as indicated by a high frequency of APs (Figure 2A–C). A morphological analysis of the biocytin-loaded GFP-expressing neurons revealed typical PV interneuron morphology, i.e., a 16–18 μm soma diameter with aspiny dendrites and an extremely dense local axonal plexus [21] (Figure 2D).

### 2.3. Whole-Cell Recording and Biocytin Labeling of MSNs

The striatum is the main input structure of the basal ganglia and is comprised of 95% GABAergic MSNs, which integrate glutamatergic inputs from the cortex and thalamus and direct GABAergic output to downstream basal ganglia nuclei [22]. The MSNs can be recognized during a recording by their morphology, having somata that are small in diameter that appear as elliptical homogenous structures [23] under the IR-DIC microscope, and by their electrophysiological properties, such as low input resistance, strong inward rectification, long latency to initial spiking and regular firing [24]. APs of the whole-cell recorded MSNs are shown in Figure 3A–C, while the analysis of the fixed brain slices allowed the biocytin signal to be visualized (Figure 3D). The recorded neurons displayed typical MSN characteristics, such as a small soma size (10–15 μm in diameter) and large and extensive dendritic trees covered with spines.

### 2.4. Whole-Cell Recording and Biocytin Labeling of PNs

PNs constitute around 80% of the neuronal population of the brain. PNs in layer V of the M1 region are projection neurons that send axons to the striatum, the midbrain, and the spinal cord [25]; they play a critical role in motor learning, for example in the calibration and acquisition of motor skills [26] and in high-level cognitive decisions [27]. We next recorded and labeled the PNs in the acute brain slices. Under the IR-DIC microscope, PNs were recognized by their teardrop pyramidal cell bodies [6]. After the establishment of whole-cell mode, APs of the PNs were captured and analyzed offline. The APs of PNs have been classified as regular- and burst-firing types [28]; we observed regular-pattern firing here (Figure 4A–C). The analysis of the biocytin signals in the fixed brain slices revealed morphological properties typical of PNs, such as a cluster of short basal dendrites emerging from the rounded end of the soma, and apical dendrites emerging from the pointed end of the soma (16–21 μm in diameter) [29,30] (Figure 4D).

### 2.5. Analysis of Biocytin-Labeled Neurons by ImageJ

The structure and dynamics of spines are associated with synapse strength and plasticity, both of which are severely impacted in various neurological diseases [31]. After whole-cell recording, we obtained brain slices in which the recorded neurons were labeled with biocytin. The slices were fixed and further prepared for immunohistochemistry, as indicated in Methods. Next, we used the freely available software ImageJ to capture and analyze images of various biocytin-labeled neurons (Appendix A) to analyze the dendritic length, number of intersections and spine density of the biocytin-labeled neurons. Together, the results suggest that the combination of whole-cell patch-clamp recording with biocytin labeling is a powerful tool for understanding single-neuron structure–function correlations.

### 2.6. Altered APs and Spine Density in PNs of the M1 Cortical Region of Cyld^−/−^ Mice

CYLD, which regulates dendritic arbor and spine development in PNs [15], is highly expressed in the brain [32]. We have previously shown that CYLD modulates neuronal activity in the PNs of basolateral amygdala (BLA) and in the MSNs of the striatum [33,34] and is critically involved in mediating anxiety-like behavior via activation of striatal microglia [35]. We next evaluated whether CYLD plays a role in determining the properties of APs and spine morphology in PNs of the M1. Recordings (Figure 5A,B) showed a significantly increased number of APs in the PNs of the *Cyld*^−/−^ mice, which suggests increased excitability (Figure 5C, interaction between genotype and current injection, F_(8, 185)_ = 1.330, *p* = 0.231, main effect of genotype, F_(1, 185)_ = 22.11, *p* < 0.0001; main effect of current, F_(8, 234)_ = 55.21, *p* < 0.0001, two-way repeated measures ANOVA). Meanwhile, rise time of the AP increased significantly (Figure 5D, *Cyld^+/+^*: 0.608 ± 0.031, *Cyld*^−/−^: 0.810 ± 0.048, t = 3.679, *p* = 0.0006, unpaired Student’s *t*-test) while leaving decay time unchanged in the *Cyld*^−/−^ mice (Figure 5E, *Cyld^+/+^*: 2.818 ± 0.218, *Cyld*^−/−^: 2.732 ± 0.180, t = 0.296, *p* = 0.773, unpaired Student’s *t*-test). We then assessed the morphology of the PNs in the *Cyld^+/+^* and *Cyld*^−/−^ mice and measured the complexity of dendritic arborization (Figure 5G–I). The results showed that, although dendritic length (Figure 5J, *Cyld^+/+^*: 6.017 ± 0.411, *Cyld*^−/−^: 5.694 ± 0.365, t = 0.620, *p* = 0.5448, unpaired Student’s *t*-test;) and number of intersections (Figure 5K, interaction between genotype and distance, F_(29, 406)_ = 2.730, *p* < 0.0001; main effect of genotype, F_(1, 14)_ = 0.0002, *p* = 0.989, two-way repeated measures ANOVA) were similar in the *Cyld^+/+^* and *Cyld*^−/−^ mice, the PNs in the *Cyld*^−/−^ mice showed a significant loss of basal and apical dendritic spines as indicated by a significantly decreased density of basal dendritic spines (Figure 5L, *Cyld^+/+^*: 0.6614 ± 0.038, *Cyld*^−/−^: 0.500 ± 0.029, t = 3.204, *p* = 0.003, unpaired Student’s *t*-test) and apical dendritic spines (Figure 5M, *Cyld^+/+^*: 0.713 ± 0.066, *Cyld*^−/−^: 0.504 ± 0.030, t = 3.209, *p* = 0.002, unpaired Student’s *t*-test). Considering that spine density is negatively corelated with input resistance (R_in_) [36] and R_in_ is positively corelated with neuronal excitability [36,37,38], we assessed the R_in_ of the PNs and observed a significantly increased R_in_ in the *Cyld*^−/−^ PNs (Figure 5F, *Cyld^+/+^*: 205.807 ± 15.016, *Cyld*^−/−^: 267.181 ± 22.168, t = 2.358, *p* = 0.023, unpaired Student’s *t*-test).

After hyperpolarization (AHP) measures the hyperpolarizing phase of action potential, which is a feedback mechanism for regulating the patterning neuronal firing [39,40]. We observed a significantly decreased AHP duration and amplitude and an increased AHP decay in the *Cyld*^−/−^ mice (Figure 6A–C, *Cyld ^+/+^*: 0.066 ± 0.006, *Cyld*^−/−^: 0.038 ± 0.004, t = 3.788, *p* = 0.0004, unpaired Student’s *t*-test; Figure 6D, *Cyld^+/+^*: 9.471 ± 0.534, *Cyld*^−/−^: 7.481 ± 0.353, t = 2.869 *p* = 0.006, unpaired Student’s *t*-test; Figure 6E, *Cyld^+/+^*: 231.077 ± 20.816, *Cyld*^−/−^: 306.997 ± 32.510, t = 2.055, *p* = 0.046, unpaired Student’s *t*-test).

## 3. Discussion

It is widely accepted that electrophysiology involves some of the most challenging experimental techniques, (e.g., patch-clamp recording) in neuroscience laboratory work. To achieve successful whole-cell recordings, the viability of acute brain slices is key. A low-temperature cutting environment, which includes the cutting solution and cutting chamber as well as the speed of cutting, are critical. The correct thickness of the brain slices should also be considered. In this study, the slice thickness was 320 μm, which ensured quality work for both whole-cell recording and after-recording post-fixation slicing. The biocytin signal was visualized using either Streptavidin Alexa 594 (red) or Streptavidin Alexa 488 (green). It is worth noting that when the slice includes a fluorophore resulting from treatment with a viral expression construct, such as GFP expressed in PV interneurons, biocytin should be developed using the fluorescence of a different color, e.g., red, as shown in the present study.

As for the electrode used for both whole-cell patch-clamp recording and biocytin diffusion, the tip of the glass capillary should have a high resistance for neurons with small somata but a low resistance for neurons with large somata. Another important factor is that there should be no bubbles in the tip when filling the electrode. Moreover, when the recording of the APs is complete, the electrode should be maintained in whole-cell mode for at least 40 min to ensure sufficient diffusion of the biocytin into the cell. A common issue seen in biocytin-labeled cells is that the fluorescent signal appears inhomogeneous, which can impair a detailed analysis of neuronal morphology, particularly for dendritic spines. We found that such an inhomogeneous biocytin signal is mainly due to the following reasons: (1) insufficient biocytin diffusion time; (2) the tip size of the patch pipette is inappropriate such that resealing after biocytin diffusion is not achieved; (3) the labeled neuron is in poor condition or, following biocytin loading, is damaged or has detached after retraction of the pipette. However, the detailed methods introduced in this article should help researchers to master the combined technique quickly.

We simultaneously conducted whole-cell recording and the biocytin-labeling technique with the PV neurons, MSNs and PNs of the corticostriatal network in the present study. Corticostriatal connections are critical for integrating information for movement control, habit formation, reward-seeking and cognitive function [41,42,43]. PNs, MSNs and PV interneurons are important for the processing of corticostriatal neuronal network activity [44,45,46,47]. The morphology and relative location of neurons largely determines the synaptic connectivity between them [48], while altered corticostriatal electrophysiological and morphological properties occur in brain diseases such as amyotrophic lateral sclerosis (ALS) and Huntington’s and Parkinson’s diseases [49,50,51]. Therefore, revealing neuronal electrophysiological and morphological characteristics in addition to molecular signaling pathways is key for improving our understanding of physiological activity and for finding targets to treat the abnormalities that occur in neuropsychiatric and motor disorders.

The *Cyld* gene is involved in various neurological disorders, including anxiety [12,35], frontotemporal dementia, and ALS [52,53,54,55], and CYLD is widely distributed in the brain, with its highest expression level in the striatum [56]. In line with findings showing that glutamate-induced excitotoxicity with hyperexcitation and spine loss in PNs are common in various pathological conditions [57,58], we observed a significantly increased firing of the APs in the PNs associating with enhanced R_in_ and reduced basal and apical spine density in the M1 of the *Cyld*^−/−^ mice. The results implicate an in-built compensatory mechanism in CYLD-deficient mice, of which structural changes (spine loss) caused elevated R_in_ and excitability in PNs [36]. It is worth noting that the finding of an increased number of APs in the PN of M1 differs from what we have observed in the BLA, where a reduced intrinsic excitability of the principal neurons occurred in the absence of CYLD [33]. Given that we observed, in the present study, a significantly increased R_in_, rise time of Aps, and decreased after the hyperpolarization (AHP) of the M1 PNs, which did not occur in the BLA [33], CYLD may regulate neuronal intrinsic property in a brain area and neuronal type specific manner. Our study uncovers a physiological and anatomical role for CYLD in mediating normal electrophysiological and dendritic spine properties in PNs.

Single-cell labeling combined with whole-cell recording is an excellent technique for investigating correlations between the physiological and morphological characteristics of the same neurons, thereby guiding further in-depth mechanistic research. The protocol introduced here has several advantages over other commonly used labeling techniques that are not in combination with electrophysiology, such as Golgi–Cox staining, carbocyanine dye loading, and AAV-sparse labeling, etc. Golgi–Cox staining uses dyes that have a high toxicity and a low success rate, and it involves a long and complex experimental procedure. The carbocyanine dyes used to characterize neuronal morphology and track neuronal pathways, such as octadecyl-indocarbocyanine (Dil) and oxycarbocyanine (Dio) [59], cannot be used on top of whole-cell recording. Sparse labeling technology visualizes the neuronal morphology by AAV viral microinjection [60,61], while the enhanced fluorescence background induced by fluorophore is hard to solve, impairing the image quality of the morphology, especially in spine morphology. Although Lucifer Yellow (LY) can also be introduced into neurons via patch-pipettes [62], it has obvious shortcomings. LY has low solubility which restricts the concentration of LY in an intracellular solution and easily precipitates by potassium ion, resulting in a blockage of the recording electrode tip during iontophoresis [63]. In contrast, biocytin, small molecules loaded into cells through patch-pipettes, can quickly diffuse throughout the neuronal fine structure. In addition, the biocytin signals are stable for long-time storage at 4 °C and do not tend to fade quickly, particularly under a confocal microscope, resulting in high-quality images of the neuronal architecture and spine morphology. Therefore, biocytin labeling combined with electrophysiological recording has been becoming a mainstream technique in the field of electro-neurobiology [5,9,64,65]. Moreover, in combination with viral labeling as shown by the present study, biocytin can target a specific group of neurons, making it an invaluable technique in the field of neuroscience research.

## 4. Materials and Methods

### 4.1. Animals

Adult male and female C57BL/6J mice, *Pvalb-Cre* mice (a generous gift from Dr. Minmin Luo at the National Institute of Biological Sciences, Beijing, China), *Cyld^+/+^* mice and littermate *Cyld*^−/−^ mice three months of age were used. *Pvalb-Cre* mice, which express Cre recombinase under the control of a PV promoter, were maintained on a C57BL/6J background. Cre was expressed heterozygotically [18]. *Cyld^+/+^* mice and their littermate *Cyld*^−/−^ mice were generated from *Cyld^+/−^* mice (a generous gift from Dr. Shao-cong Sun at the University of Texas MD Anderson Cancer Center, Houston, TX, USA) [66]. Mice were housed under standard conditions (12 h light/dark at constant temperature). Food and water were freely available. The Chinese Ministry of Science and Technology Laboratory Animals Guidelines were followed, and experiments were approved by the local ethical committee of Guangzhou University and South China Normal University.

### 4.2. Stereotaxic Viral Injection

To specifically identify PV interneurons, *Pvalb-Cre* driver lines were combined with an adeno-associated virus (AAV) engineered to conditionally express enhanced green fluorescent protein (EGFP) in the presence of the Cre recombinase (AAV-hSyn-DIO-EGFP) (BrainVTA, Wuhan, China) [18]. AAV-hSyn-DIO-EGFP (1 × 10^12^ genome copies (GCs)/mL) should be thawed and divided into 4 μL aliquots on dry ice, after which it can be stored at −80 °C for several years. In this protocol, we used 6–8-week-old *Pvalb-Cre* mice for viral injections.

Virus injection: Sterilize the surgical tools and disinfect the surgical equipment using 70% (*v*/*v*) ethanol. Pull virus-injection pipettes to a tip diameter of 20 μm and slowly fill the pipette with paraffin oil. Thaw an aliquot of AAV-hSyn-DIO-EGFP, dilute with sterile 1× phosphate buffered saline (PBS) to give a viral titer of 5 × 10^11^ GCs/mL, and keep it on ice. Anesthetize the mouse by inhalation of 3% isoflurane and then place the animal in the stereotaxic apparatus. Use a heated blanket to maintain the mouse’s body temperature and administer 1.5% isoflurane until injection of the virus is complete. Inject carprofen (5 µg/g), a pain-killing and anti-inflammatory medicine, subcutaneously using a sterile single-use syringe. Stabilize the mouse’s head in the stereotactic frame (RWD Life Science, Shenzhen, China), and use erythromycin ointment to protect the eyes. Shave the hair on the mouse’s head and disinfect the scalp with iodine solution followed by 70% (*v*/*v*) ethanol. Use scissors and forceps to make a midline incision, expose the skull, and then position a dissecting microscope (Zeiss, Oberkochen, Germany). Add a few drops of sterile normal saline to clean the skull, and dry the area with cotton swabs. Confirm the location of the bregma under the dissecting microscope and mark the area of the dorsolateral striatum with a pen. The stereotactic coordinates used for the dorsolateral striatum are anteroposterior, 0.74 mm; dorsoventral, −2.07 mm; mediolateral, ±2.27 mm; and at a 6° angle. Drill slowly and carefully into the skull, using sterile saline to avoid overheating of the drill tip, until a small hole is obtained in the desired area. Install a 1-μL syringe (Hamilton Co., Bonaduz, Switzerland) with a virus-injection pipette into a manipulator, and draw the virus solution into the pipette. Inject 300 nL at a rate of 30 nL/min into each hemisphere of the targeted region. Following injection, allow the virus to diffuse into the injection site for an additional 10 min, and then slowly withdraw the virus-injection pipette. Stitch the wound with sterile suture line. Place the mouse in a recovery cage (length 60 cm, width 30 cm, height 20 cm) at 37 °C until it recovers consciousness. Then, return the animal to its home cage for at least three weeks before patch-clamp recording and dye loading experiments are performed.

### 4.3. Internal and External Solutions for Patch-Clamp Recording

#### 4.3.1. Artificial Cerebrospinal Fluids (aCSFs)

Prepare 500 mL slicing aCSF containing the following (in mM): 93 N-methyl-D-glucamine, 2.5 KCl, 1.2 Na_2_HPO_4_, 30 NaHCO_3_, 20 HEPES, 25 D-glucose, 5 sodium L-ascorbate, 2 thiourea, 3 sodium pyruvate, 10 MgSO_4_, 0.5 CaCl_2_. Prepare 250 mL incubating aCSF containing the following (in mM): 92 NaCl, 2.5 KCl, 1.2 NaH_2_PO_4_, 30 NaHCO_3_, 20 HEPES, 25 D-glucose, 5 sodium L-ascorbate, 2 thiourea, 3 sodium pyruvate, 2 CaCl_2_, and 2 MgSO_4_. Prepare 250 mL extracellular recording aCSF containing (in mM) 124 NaCl, 2.5 KCl, 2 CaCl_2_, 1.2 NaH_2_PO_4_, 24 NaHCO_3_, 2 MgSO_4_, 12.5 D-glucose, 5 HEPES. Saturate the aCSFs with 95% O_2_/5% CO_2_ for at least 15 min, and adjust to pH 7.3 using 5 M HCl [67].

#### 4.3.2. Intracellular Biocytin Solution

The following solution and aliquots should be prepared and kept on ice to prevent ATP/GTP degradation. Prepare 25 mL internal solution containing (in mM) 0.1% biocytin (B4261, Sigma, St. Louis, MO, USA), 110 K-gluconic acid, 10 NaCl, 1 MgCl_2_, 10 EGTA, 40 HEPES, 2 Mg-ATP, 0.3 Na-GTP. Adjust to pH 7.3 using 5 M NaOH; the osmolarity is 280–300 mOsm.

### 4.4. Slice Preparation for Whole-Cell Patch-Clamp Recording

High-quality slice preparation is critical for successful patch-clamp recording. Anesthetize mice with 2.5% chloral hydrate (0.15 mL/10 g) and perfuse with prechilled and oxygenated slicing aCSF. After decapitation, hold the head and cut the skin along the midline with surgical scissors, exposing the skull. Using small straight sharp-tipped scissors, make lateral cuts in the skull posterior and a cut between the eyes, then cut carefully along the sagittal suture and peel off the skull.

Conduct this process quickly (<60 s) to avoid anoxic shock to the neurons. Gently remove the brain from the skull and immediately place it in ice-cold slicing aCSF bubbled with 95% O_2_/5% CO_2_ for 1 min to cool. Place the brain on top of the filter paper in the ice-filled glass petri dish and add ice-cold aCSF onto the brain. Use a precooled razor blade to trim the brain and obtain the desired coronal plane. Use a vibratome to cut 320 μm coronal slices containing the primary motor cortex (M1) and striatum. Glue brain tissue onto the specimen-holding plate and, when the glue is dry, secure the plate in the ice-cold aCSF-filled slicing chamber, and continuously gas with 95% O_2_/5% CO_2_ until slicing is complete. Set the vibratome with the following parameters: speed, 0.15 mm/s; vibration, 3500 rpm; slice thickness, 320 μm. After slicing, transfer the slices to the recovery chamber filled with slicing aCSF, and allow the slices to recover at 30 °C for 10 min. Carefully transfer the slices to the incubation chamber filled with incubation aCSF, and incubate at RT for another 1 h before recording.

### 4.5. Electrodes for Whole-Cell Recording and Biocytin Diffusion

Pull glass electrodes using an electrode puller (PC-10, Narishige, Tokyo, Japan). Adjust the tip size of the glass electrodes to the soma size of the neurons being targeted. For recording MSNs (10–15 μm soma diameter), electrodes with 6–8 MΩ resistance are required. For recording PV interneurons (somata average diameter 16–18 μm) and PNs (16–21 μm in diameter), the electrode resistance ranges from 5 to 6 MΩ. Thaw an aliquot of intracellular biocytin solution and keep it on ice to prevent ATP/GTP degradation.

Turn on the constant flow pump to deliver the recording aCSF at a flow rate of 1.5–2.5 mL/min. Continuously perfuse the solutions with 95% O_2_/5% CO_2_ throughout the recording session. Turn on the perfusion heater such that the recording aCSF flowing into the recording chamber is maintained at a stable temperature of 30 °C. Turn on all the electrical equipment.

### 4.6. Whole-Cell Recording of APs

#### 4.6.1. Selection of Target Neurons for Whole-Cell Recording

Transfer a brain slice to the recording chamber and hold the slice in place using a slice anchor made of mesh. Under the IR-DIC microscope (Nikon, Tokyo, Japan), locate the targeted region of the brain using the 4× objective, and then switch to the 40× water immersion lens to identify neurons. Healthy neurons should possess the following characteristics: first, the nucleus is not visible; second, a good neuron has a three-dimensional appearance, with a smooth and bright cell membrane. To ensure neuron integrity, do not select a cell at the surface of the brain slice; cell location is a compromise between neuron integrity and visibility. Identify cells primarily by the shape of their somata, and choose PV interneuron, MSN and PN targets for whole-cell recording. Use epifluorescence microscopy (Nikon, Japan) to identify fluorescently labeled PV interneurons of *Pvalb-Cre* mice [18].

#### 4.6.2. Giga-Ohm (GΩ) Seal Formation and Establishment of Whole-Cell Mode in the Target Neuron

Gently apply positive pressure to the solution in the electrode to avoid spilling the biocytin-containing internal solution into the extracellular space, which would result in high background staining. Lower the recording electrode into the bath, set the amplifier to voltage-clamp mode to perform the membrane test in “Bath” mode, and adjust the electrode offset current to zero. Under the microscope, find and move the electrode slowly toward the brain slice using the micromanipulator. When it is just above the cell of interest, reduce the speed to low mode to avoid puncturing the neuron with the electrode tip. Apply positive pressure to blow away tissue overlying the cell. When the electrode tip makes contact with the cell, a slight increase in resistance can be observed from the recording software. When a dimple can be seen on the cell surface, the pressure is switched from positive to weak negative. When the resistance reaches 100 Ω, switch from “Bath” to “Patch” mode in the Membrane Test. Once the resistance reaches 300 Ω, release the negative pressure, and the resistance should then reach at least 1 GΩ. Switch from “Patch” to “Cell” mode in the Membrane Test, set the holding potential as close as possible to the physiological resting membrane potential of the cell (e.g., −60 to −70 mV), and then compensate for fast and slow pipette capacitance. Apply brief, firm suction to open the plasma membrane of the cell. Once the whole-cell configuration has been established, different parameters of the cell, such as membrane capacitance (C_m_), membrane resistance (R_m_) and series resistance (R_s_), can be monitored. If Rs is larger than 25 MΩ, the cell should be discarded. Set the amplifier to current-clamp mode and bridge balance, and apply a protocol delivering a series of 50- or 1000- ms depolarizing currents to measure changes in membrane voltage. Data are acquired with a Digidata 1440A interface and pClamp 10.4 software (Molecular Devices, San Jose, CA, USA) and a MultiClamp 700B amplifier (Molecular Devices, USA). Analyze the number of APs to measure intrinsic tonic firing activity in neurons using ClampFit software (version 11.03.03, Molecular Devices, USA) (https://www.moleculardevices.com/, accessed on 7 February 2022).

### 4.7. Biocytin Diffusion and Brain-Slice Fixation

To visualize the patched neurons, it is necessary to label them by introducing a marker material, biocytin, into the intracellular solution. After recording APs, stay in whole-cell mode to allow the biocytin to diffuse out of the electrode into the recorded neuron. Neurons must be recorded for sufficient time (40–60 min) to allow for the adequate diffusion of biocytin into the fine structures of the neuron, such as the axon and spines. After biocytin diffusion, the electrode must be slowly retracted to reseal the neuron. Leave the slice in the recording solution for 10 min to wash off excess biocytin in the extracellular space. Fix the brain slices in 4% PFA in a dish at 4 °C and place on a rotating shaker for 24 h.

### 4.8. Post-Hoc Immunohistochemistry

Wash the brain slices with PBS three times, 5 min each time, at RT. Permeabilize the brain slices in 1% Triton-X100 (Sigma)/PBS at 4 °C for 24 h. Wash the slices with PBS for 5 min. Incubate the sections in Streptavidin Alexa 488 (S11223, Invitrogen, Waltham, MA, USA) or Streptavidin Alexa 594 (S11227, Invitrogen, USA) diluted 1:1000 in 0.3% Triton-X100/PBS at 4 °C for 24 h. After a final series of three washes with PBS, mount the slices on glass microscope slides. All steps of this process are performed on a rotating shaker and in the dark.

### 4.9. Image Processing

Z-stacks of neurons expressing fluorescent proteins are collected using a confocal microscope (LSM-800, Zeiss, Germany) with 20× and 63× objectives. Set up the multichannel configuration according to the properties of the fluorescent protein infused into neurons. Position the fluorescently labeled neuron in the visual field of the camera, adjust the focus, and then determine the exposure time for all channels one after another. Activate the Z-stack checkbox, manually set up the configuration parameter, including the first, last and center planes of the focus position, and the interval between planes (20× objective, 2 μm; 63× objective, 0.2 μm). Acquire the Z-stack images. After imaging, store the slides at 4 °C in a dark holder.

### 4.10. Morphological Analysis of Biocytin-Labeled Neurons

Analysis parameters include the length of dendrites, the surface area of dendrite coverage, the number of dendrite intersections at various distances from the cell body (Sholl analysis) and dendritic spine density. Analyze the sum length of dendrites. Run ImageJ (version 1.51n, NIH, Bethesda, MD, USA) (https://fiji.sc/, accessed on 18 July 2022), select “File—Import—Import Sequence” in the window, open high-quality Z-stacks of biocytin-filled neurons (20×), convert RGB image to an 8-bit luminance image. Set appropriate scale. Select “Plugins—Segmentation—Simple Neurite Tracer”, then create a new 3D viewer. Construct a tracing of a biocytin-loaded neuronal dendrite with a start point at the soma and an end point at the tip of a dendrite. When a single dendritic trace is finished, click on “Finish Path”; the color will change from red to purple. Select “Analysis—Measure Paths”, and export all dendritic length data. Select “File—Save Traces File” to save the file.

Analyze the complexity of neuronal dendritic arborization using “Sholl analysis” [68]. Mouse over the path of the dendrite of interest, press “G” to activate “Sholl analysis”. Press “Ctrl + Shift” to mouse over the neuronal soma along the path, then press “Ctrl + Shift + A” to start the analysis. Set “radius step size” to 10 μm. The radius step size is the sampling interval between the radii of consecutive sampling circles/spheres. Save the analysis profiles. Manually analyze the dendritic spine density and the spine density of second branch dendrites. Select “File—open” to open the Z-stack image of dendrites (63×). Click on the “Segmented line” icon, and mouse over the path, select “Analyze—Measure” to test the length of a target dendrite. Select “Plugins—Analyze—Cell counter—Initialize”, rename type 1 as “spine”, and click on the image to mark the spine on a single dendrite, then save the number of spines. Calculate spine density as the number of spines divided by dendritic length.

### 4.11. Statistical Analysis

All data were expressed as the mean ± standard error of the mean (SEM) and were statistically evaluated using Student’s t-test or two-way repeated measures with Sidak’s post-hoc multiple comparison test with GraphPad Prism and SPSS. *p* < 0.05 was considered significant (* *p* < 0.05, ** *p* < 0.01, *** *p* < 0.001). Mean ± SEM values, sample size, *p*-values and statistical methods are defined in the respective results and figure legends.

## 5. Conclusions

The present study introduced a detailed procedure for a biocytin-labeling technique in combination with whole-cell recording that can be used to reveal the mechanistic relationship between the electrophysiology and morphology of a neuron. Moreover, we reported here that CYLD plays a pivotal role in controlling the excitability and dendritic spine density of PNs in the M1.

## Figures and Tables

**Figure 1 molecules-28-04092-f001:**
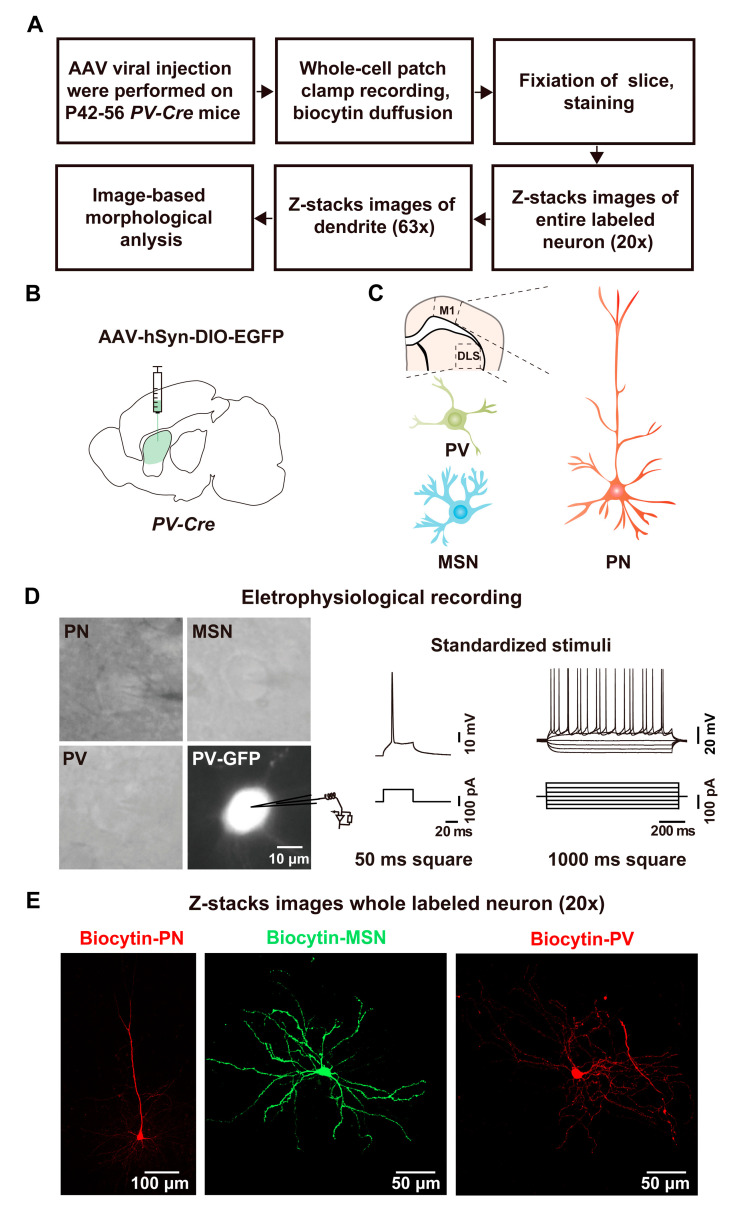
Illustration of experimental procedure. (**A**) Flow diagram of experimental procedures. (**B**) Schematic of AAV-hSyn-DIO-EGFP injection into the DLS of *PV-Cre* mice. (**C**) Schematic illustration of the areas of acute brain slices and neuron types targeted for whole-cell recording and biocytin infusion. (**D**) Neurons were initially identified by the shape of their somata using IR-DIC microscopy and were then recorded by whole-cell patch-clamping with a standard stimulation paradigm for the characterization of intrinsic electrical properties. (**E**) Slices were incubated with Streptavidin Alexa 594 (red) or Streptavidin Alexa 488 (green) for 24 h to reveal biocytin-loaded neurons under a confocal microscope. Maximum-intensity projections are shown of a biocytin-labeled PN, MSN and PV interneuron.

**Figure 2 molecules-28-04092-f002:**
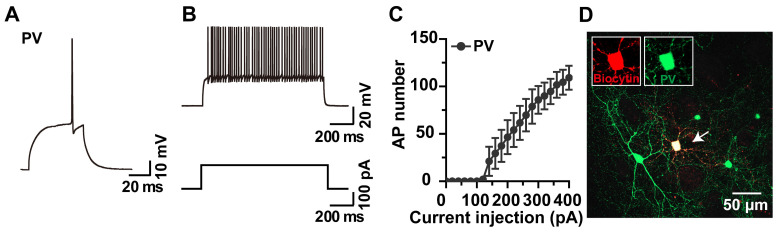
Representative traces and current-voltage curve of whole-cell recorded APs of PV interneurons and biocytin labeling of the same cells. (**A**,**B**) Representative single and current injection-induced APs of a PV interneuron. (**C**) Current-voltage (I-V) curve of PV interneurons (n = 9 neurons). (**D**) EGFP-positive PV neuron labeled with biocytin (red) to reveal its morphology (white arrow).

**Figure 3 molecules-28-04092-f003:**
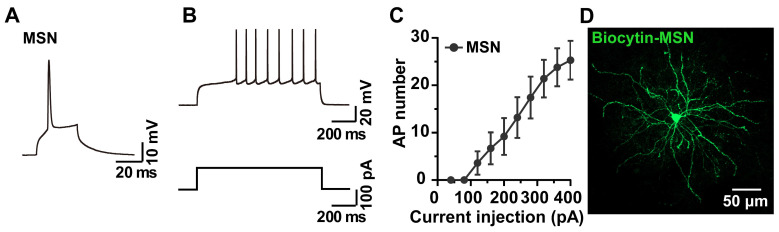
Representative traces and I-V curve of whole-cell recorded APs of MSNs and biocytin labeling of the same cells. (**A**,**B**) Representative single and current injection-induced APs of an MSN. (**C**) I-V curve of MSNs (n = 10 neurons). (**D**) Morphology of an MSN revealed by biocytin.

**Figure 4 molecules-28-04092-f004:**
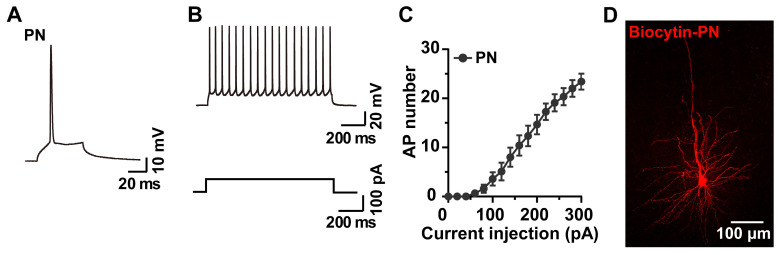
Representative traces and I-V curve of whole-cell recorded APs of PNs and biocytin labeling of the same cells. (**A**,**B**) Representative single and current injection-induced APs of a PN. (**C**) I-V curve of PNs (n = 12 neurons). (**D**) PN morphology revealed by biocytin.

**Figure 5 molecules-28-04092-f005:**
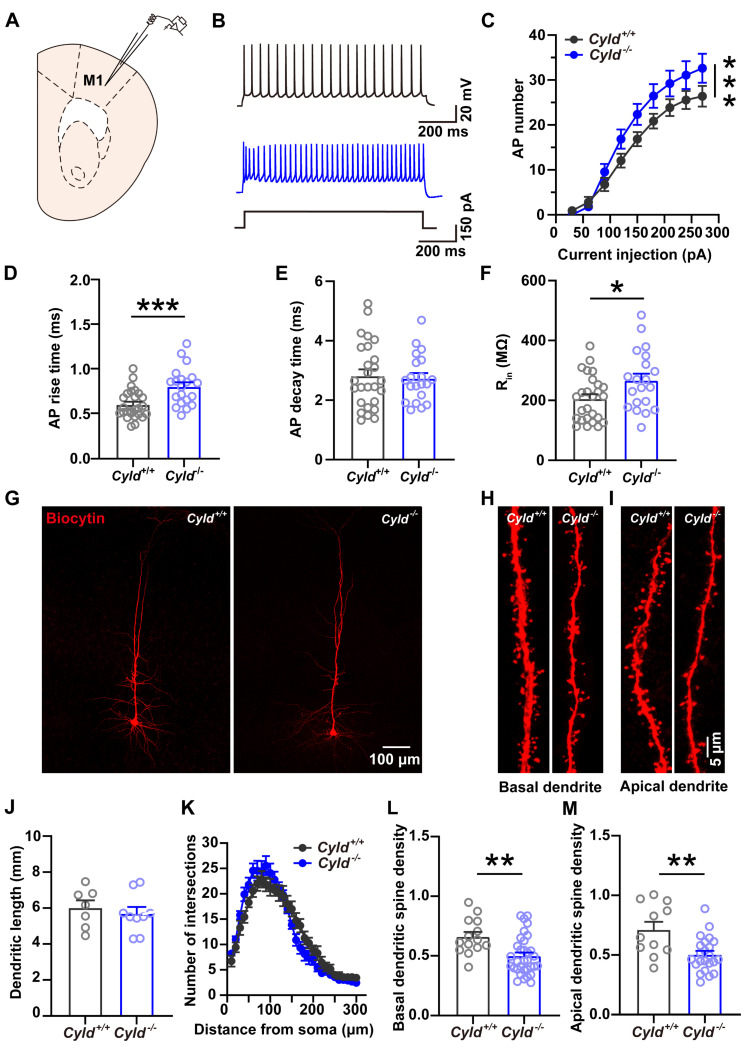
Apical and basal dendritic spine loss in PNs of *Cyld*^−/−^ mice. (**A**) Diagram of the electrophysiological recording strategy in acute motor cortical slices. (**B**) Representative traces of AP firing of PNs of *Cyld^+/+^* mice (black) and *Cyld*^−/−^ mice (blue) in response to 150 pA (1000 ms duration) current injections. (**C**) Depolarizing current steps (+30 to +270 pA, 30 steps, 1000 ms duration) lead to an increase in AP frequency in *Cyld*^−/−^ mice (n = 27 neurons from 11 *Cyld^+/+^* mice, n = 22 neurons from 7 *Cyld*^−/−^ mice). (**D**–**F**) Bar graphs showing AP rise (**D**), decay time (**E**) and R_in_ (**F**) of PNs from *Cyld^+/+^* and *Cyld*^−/−^ mice (n = 26 neurons from 11 *Cyld^+/+^* mice, n = 20 neurons from 7 *Cyld*^−/−^ mice). (**G**) Representative image of an individual PN from *Cyld^+/+^* or *Cyld*^−/−^ mice loaded with biocytin via a patch microelectrode. PNs were imaged with a confocal microscope system, and images were used to analyze dendrite morphology and spine density. (**H**,**I**) Representative confocal images of basal dendrites (**H**) and apical dendrites (**I**) from *Cyld^+/+^* and *Cyld*^−/−^ mice. (**J**) Bar graphs showing dendritic length of PNs from *Cyld^+/+^* and *Cyld*^−/−^ mice. (**K**) Sholl analysis demonstrating that there is no difference between the genotypes in the neuronal dendritic arborization complexity of PNs. Points represent individual neurons (n = 7 neurons from three *Cyld^+/+^* mice, n = 9 neurons from three *Cyld*^−/−^ mice) in (**J**,**K**). (**L**,**M**) Bar graphs showing that basal (**L**) and apical (**M**) dendritic spine density of PNs decrease in *Cyld*^−/−^ mice. Points represent single basal dendrites (n = 14 dendrites from *Cyld^+/+^*, n = 33 dendrites from *Cyld*^−/−^) in (**L**) and single apical dendrites (n = 11 dendrites from *Cyld^+/+^*, n = 23 dendrites from *Cyld*^−/−^) in (**M**). Data are presented as the mean ± SEM; * *p* < 0.05, ** *p* < 0.01, *** *p* < 0.001.

**Figure 6 molecules-28-04092-f006:**
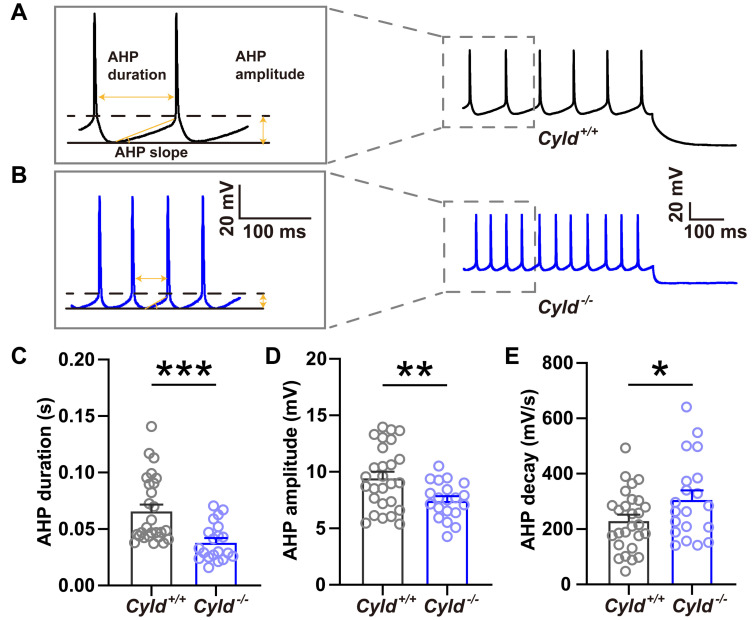
Decreased AHP duration and increased AHP decay in PNs of *Cyld*^−/−^ mice. (**A**,**B**) Representative traces of AP firing of PNs in *Cyld^+/+^* (**A**) and *Cyld*^−/−^ mice (**B**) in response to 150 pA (1000 ms duration) current injection. Insets show expanded view of the spikes. (**C**–**E**) Bar graphs of AHP duration (**C**), amplitude (**D**) and decay constant (**E**) (n = 29 neurons from 10 *Cyld^+/+^* mice, n = 20 neurons from 5 *Cyld*^−/−^ mice). Data are presented as the mean ± SEM; * *p* < 0.05, ** *p* < 0.01, *** *p* < 0.001.

## Data Availability

All data generated or analyzed in this study are included in this article.

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
