# Peer review of "Biocytin-Labeling in Whole-Cell Recording: Electrophysiological and Morphological Properties of Pyramidal Neurons in CYLD-Deficient Mice"

_molecules, 2023, doi:10.3390/molecules28104092_

Round 1

Reviewer 1 Report

Shu-yi Tan et.al paper entitled " Biocytin-labeling in whole-cell recording: electrophysiological and morphological properties of pyramidal neurons in CYLD-deficient mice” is a methodological paper. I have few concerns regarding the paper.

Can the authors show that there were no morphological changes in neurons before and after completing the whole process of recording and labelling? I see that AAV virus expresses EGFP and those figures colocalizing with Biocytin is missing. I will like the authors to include that panel also.

Minor concerns:

1.     Please change on of the images as Fig1E and Fig 4D is same. Include other pictures.

2.     Screenshots of ImageJ (fig 5A-C) panel could be moved into the supplement and start fig5 with D. The detail of method is already there is the methods section no need to provide screenshot of the software.

3.     Fig 6F, WT mice images has less dendritic arborization than the KO mice. Can the authors change the image and select the suitable one for representation based on their analysis.

The quality of English can be improved and is average.

Reviewer 2 Report

The first part, comprising of AAV-mediated GFP tagging, whole-cell patch, biocytin-streptavidin histology, and morphometric analyses, is fairly standard and practiced well by the authors. While this is OK, it is rather confusing to put MSN data since the second part of the manuscript is about PNs in CTKD-KO. The authors might consider putting the first part as a separate protocol manuscript. The second part of the paper reports that M1 L5 PNs in CYLD-KO is more excitable while the spine density is lower. Overall, this might mean that the input resistance of CYLD-KO PNs is higher than wild type. It is interesting that this phenotype is not observed in the BLA in a prior study (referenced as [33]). No mechanism is discussed.

Round 2

Reviewer 1 Report

All my comments were addressed. I have no further concerns.